# Characterization of Food Packaging Films with Blackcurrant Fruit Waste as a Source of Antioxidant and Color Sensing Intelligent Material

**DOI:** 10.3390/molecules26092569

**Published:** 2021-04-28

**Authors:** Mia Kurek, Nasreddine Benbettaieb, Mario Ščetar, Eliot Chaudy, Maja Repajić, Damir Klepac, Srećko Valić, Frédéric Debeaufort, Kata Galić

**Affiliations:** 1Faculty of Food Technology and Biotechnology, Pierottijeva 6, 10000 Zagreb, Croatia; mscetar@pbf.hr (M.Š.); maja.repajic@pbf.unizg.hr (M.R.); kgalic@pbf.hr (K.G.); 2Department of BioEngineering, Institute of Technology, University of Burgundy, 7 Blvd Docteur Petitjean, BP 17867, CEDEX, F-21078 Dijon, France; nasreddine.benbettaieb@u-bourgogne.fr (N.B.); chaudy.eliot1@gmail.com (E.C.); frederic.debeaufort@u-bourgogne.fr (F.D.); 3UMR PAM A 02.102, Agrosup Dijon, University of Bourgogne Franche-Comté, F-21000 Dijon, France; 4Faculty of Medicine, University of Rijeka, Braće Branchetta 20, 51000 Rijeka, Croatia; damir.klepac@medri.uniri.hr (D.K.); svalic@medri.uniri.hr (S.V.); 5Centre for Micro- and Nanosciences and Technologies, University of Rijeka, Radmile Matejčić 2, 51000 Rijeka, Croatia

**Keywords:** blackcurrant waste, chitosan, pectin, antioxidant, intelligent sensing, color changing, packaging films

## Abstract

Chitosan and pectin films were enriched with blackcurrant pomace powder (10 and 20% (*w*/*w*)), as bio-based material, to minimize food production losses and to increase the functional properties of produced films aimed at food coatings and wrappers. Water vapor permeability of active films increased up to 25%, moisture content for 27% in pectin-based ones, but water solubility was not significantly modified. Mechanical properties (tensile strength, elongation at break and Young’s modulus) were mainly decreased due to the residual insoluble particles present in blackcurrant waste. FTIR analysis showed no significant changes between the film samples. The degradation temperatures, determined by DSC, were reduced by 18 °C for chitosan-based samples and of 32 °C lower for the pectin-based samples with blackcurrant powder, indicating a disturbance in polymer stability. The antioxidant activity of active films was increased up to 30-fold. Lightness and redness of dry films significantly changed depending on the polymer type. Significant color changes, especially in chitosan film formulations, were observed after exposure to different pH buffers. This effect is further explored in formulations that were used as color change indicators for intelligent biopackaging.

## 1. Introduction

The development of novel food packaging with ecological character is rapidly increasing. Huge efforts are being made to reduce food waste and to close the product life cycle by extending the shelf life of food, using its waste for other processing and reusing primary or secondary packaging, either for food contact or for other purposes. Moreover, there is great scientific evidence in the application of various sources of food waste for the production of food packaging materials, using various natural biopolymers from the food waste industry. Accordingly, with the development of new extraction methods, e.g., microwave assisted extraction (MAE), ultrasound assisted extraction (UAE), pulsed electric extraction (PEE), accelerated solvent extraction, which are described as natural and consumer friendly, and various fruit pomace have been investigated as sources of natural pigments, dyes, antioxidants and antimicrobials. In this respect, experiments have been undertaken in their application as a source of active ingredients or functional packaging. The term “functional” refers to either an active or intelligent character, such as an antioxidant film that stops or slows the oxidation of packaged fatty products, or an intelligent sensor capable of detecting the presence of progressively formed volatile amines in spoiled foods. These include conductometric [1] and pH sensitive components incorporated into the packaging material as carriers [2]. Antioxidant compounds could scavenge the free radicals and prevent the degradation of the packed food, resulting in longer shelf-life. In dark fruits, the abundant color pigments, polyphenols, are mainly responsible for the antioxidant effect. At the same time, these pigments can be used as pH sensitive colorants, so that the final packaging can be equipped with active antioxidants and intelligent color recognition functions. 

Since fruits are valuable foods that are mostly used fresh or processed into various products, commercial use for antioxidant extraction is not very likely. Nevertheless, by-products are great sources of the functional ingredients, although unfortunately they are often considered waste. In recent years, this aspect has increasingly changed. For example, fruit pomace, defined as leftovers after fruit processing and juicing, is increasingly being used as a source of valuable ingredients. The idea of reusing pomace has been explored in the scientific literature, but there are few references to the use of blackcurrant (*Ribes Nigrum*). This fruit is also known as cassis and it has its natural origin and is cultivated in central–northern Europe. In the Burgundy region, it is known as a typical French product and is used either in Cassis de Dijon, as a liqueur, as berry juice or as a flavor enhancer for typical French products. Blackcurrant is rich in polyphenols with more than 90% anthocyanins (e.g. delphinidin-3-*O*-glucoside, delphinidin-3-*O*-rutinoside, cyanidin-3-*O*-glucoside and cyanidin-3-*O*-rutinoside [3]). New sustainable developments are focused on the utilization of by-products for further functional applications due to the reuse of the high presence of bioactive compounds [4,5]. An interesting study was recently conducted on the valorization of processed berry waste in cereal-based foods to improve the nutritional profile of novel products [6]. Similarly, blackcurrant pomace, when added as a flour substitute in the specific type of bread, can prevent the hyperglycemic effect [7]. 

Sustainability is a promising topic for the food packaging sector, which can be attractive from both end-user and the technologist perspectives. Blackcurrant waste powder (BCW) has been used as an active ingredient source for the production of bio-based films. This type of film could potentially be used as a coating for products that do not have a monophasic structure, such as biscuits [8], candies [9], bread buns [10], etc. Pectin (PEC) and chitosan (CS) are both classified as edible and biodegradable polysaccharides with a partially crystalline structure. They are also inexpensive and abundant natural polymers that use polymers to produce food-grade pectin or chitosan raw materials and has been well reported in the scientific literature [11,12]. When used as carriers of functional compounds, their intrinsic properties may change, leading to changes in barrier, thermal or mechanical film properties [13]. Therefore, after the incorporation of novel compounds, it is advisable to evaluate all the parameters (such as antioxidant and antimicrobial efficacy, etc.) that are important for the final performance and potential use in food packaging. 

The aim of this study was to analyze how the incorporation of lyophilized and ground blackcurrant pomace powder affects the physical and chemical (color, thickness, and water vapor permeability), mechanical (tensile strength, Young’s modulus, and elongation at break), thermal (differential scanning calorimetry) and surface (spectroscopy) of pectin and chitosan-based films. In addition, investigations were carried out on the antioxidant character, as well as on color changes after exposure to different pH values, which should provide information on the possible use as active or intelligent (indicator) packaging films. 

## 2. Results and Discussion

### 2.1. Physical, Chemical and Mechanical Characterization of Films

The physico-chemical properties of the tested samples are listed in Table 1. Only the PEC films were twice as high as the thickness values when BCW was incorporated. The addition of BCW possibly disturbed the PEC network, which retracted less during drying. Consequently, the water vapor permeability (WVP) was slightly higher for the PEC films with BCW compared to the plain PEC films, which is likely related to the higher moisture content in these samples and the thickness effect. The results for PEC films are in an order of magnitude higher than those reported in Reference [14], but similar to those reported in Reference [13]. In contrast, the WVP is affected differently by the incorporation of BCW in the CS. Indeed, the addition of lower concentrations of BCW resulted in a decrease in WVP, while higher concentrations did not result in significant changes in the CS samples. 

In general, moisture content (MC) and water solubility (WS) provide information about the affinity of polymers for water. Knowledge of these parameters is of great importance for foods with high moisture content when moisture content becomes a limiting factor for the shelf-life. CS films had a solubility of about 30% (Table 1). Similar results for pure CS films have been reported previously [15]. As expected, all PEC-based samples were 100% soluble in water. The differences between CS and the PEC samples could be due to their structure resulting from different interactions between the polymer chains. Depending on the application, lower solubility would be desirable for those foods where some protection from moisture is required. However, higher solubility could play an important role in the organoleptic properties of those foods where the film is applied as a coating and is intended to be consumed with the product as a whole. Accordingly, films with higher solubility (PEC) had lower moisture content.

Various parameters describing the mechanical properties of the freshly prepared dry films are given in Table 2. This analysis was performed to evaluate whether the addition of blackcurrant powder has an important effect on the mechanical behavior of PEC and CS films. In order to produce coated food products of the desired quality and uniformity, it is important that edible films and coatings have adequate integrity, good resistance to cracking and sufficient elasticity. Pure CS films had about a 17% higher elongation at break than PEC, while two other mechanical parameters did not differ significantly. CS-based samples basically had higher flexibility, probably due to the existence of β-(1-4)-d-glucosamine bonds, which are not present in PEC polymer chains [16]. Moreover, CS networks are stabilized by hydrogen bonds, hydrophobic interactions, as well as electrostatic interactions, depending on the pH. All this contributes to a higher TS and YM, and consequently to a lower deformability (lower E%). Incorporation of BCW led to a decrease in the elasticity of CS films, even up to fivefold at a higher BCW concentration, while TS values were increased. This behavior was not surprising, considering that the powders used are not completely soluble. Indeed, on the one hand, the solid particles induced discontinuities in the network that created preferential fault zones explaining a lower E. On the other hand, the BCW composition probably allows many different types of interactions with the CS and could thus strengthen the tensile strength and Young’s modulus of the film [17]. In the case of the PEC-based film, TS, YM and E decreased, although to a lesser extent. This was attributed to the discontinuities formed in the film structure and the lack of cohesion between the polymer chains after the incorporation of BCW in PEC. Similar TS results for PEC with different fruit extracts (acerola, cashew apple, papaya, pequi, and strawberry) were reported by others [14]. 

However, for all the films, the incorporation of BCW is mainly due to the physical effect of the solid particles rather than molecular interactions, as no significant change was indicated in the FTIR spectra as follows.

### 2.2. Molecular Interactions (FTIR-ATR)

Figure 1 shows the infrared spectra of CS and PEC-based films. The band assignments are given in Appendix A. No specific interactions between the main polymer chain occurred in the formulations with BCW, as there are no significant changes in the spectral data for CS formulations. All spectral data, for both CS and PEC, showed OH-stretching bands in the range of 3600–2900 cm^−1^. In CS samples, the presence of the residues of acetic acid was mainly indicated by peaks at 1410 and 1560 cm^−1^. The peak at 2929 cm^−1^ was assigned to the asymmetric stretching of the aliphatic –CH_2_ groups. Peaks at 1650 and 1546 cm^−1^ were assigned to amide I and amide II, and below 1150 cm^−1^ to symmetric stretching of C-O-C and the amino group at the C_2_ position of a pyranose ring [18,19].

No obvious changes in FTIR spectra, nor new significant peaks appeared, as there is a great similarity in the wavelengths with spectral bands of blackcurrant itself and those of the matrices. It has previously been reported that blackcurrant extract showed absorption bands in the same regions as CS, which is at the 2900, 1730, and 1600–800 cm^−1^ regions, associated with stretching, bending and deformation vibrations of components, such as cyanidin-3-*O*-glucoside, cyanidin-3-*O*-rutinoside, delphinidin-3-*O*-glucoside and delphinidin-3-*O*-rutinoside [20,21]. 

In the PEC samples, the CH-stretching bands of the methyl group of the methyl ester appeared at 2934 cm^−1^. Vibrations at 1638 cm^−1^ were attributed to the asymmetric and to symmetric vibrations of the carboxyl group (C=C stretching band) [13]. The ester carbonyl bands (-COO-pectin) were seen between 1740 and 1744 cm^−1^ and C-O single stretching bands at 1014 and 1015 cm^−1^. In PEC-BCW samples, the COOH spectral band decreased compared to PEC, indicating possible interactions of BCW polyphenols with polymer chains.

### 2.3. Appearance and Color Characterization

Measured color properties (*L**, *a**, *b**) and Δ*E* are given in Table 3. The color changes after exposure to different pH values are shown in Figure 2. Pure films did not show any visual defects such as cracks. A distinct coloration and the presence of agglomerated particles in films with BCW could be detected by the human eye (Appendix A). This appearance could be explained by the composition of the added powder, which has a very low solubility in the film-forming suspension, and by the powder particle size. Consequently, the Δ*E* values of all films with BCW were >3. Pure CS films were yellow-greenish with negative *a** and *b** values, similar to those previously reported for the same film type [22,23]. CS films with BCW had a bluish hue with negative *a** and *b** values. The coloration also affected the film lightness (*L**), which decreased significantly with increasing BCW concentration. The decrease in lightness may have a positive effect when the films are used to pack (wrap) foods that are sensitive to photooxidation. Similar evidence has been reported by others for PEC and CS [13,24,25]. 

The control PEC films (without extracts) were lighter than the control CS films. They also had positive *a** and *b** values. PEC films had *a** values close to zero, while those with extracts were positive. This was significantly changed compared to the CS films, indicating a predominance of a red cast. Moreover, the different pH of the film-forming dispersion could cause a structural alteration of the natural pigments (anthocyanins) present in BC. Indeed, color differences were observed in the preparation of FFS, depending on PEC or CS, for the same extract addition. Moreover, after solubilization of the dry films, the pH of the obtained solutions was more acidic (pH < 5) for the CS films than for the PEC-based ones (pH~6).

The evaluation of the color response of films exposed to different pH was performed after film samples were immersed in the pH buffers (ranging from pH 2 to 12). While some films appeared to be visually destroyed after immersion, this was mainly due to their solubility in water. Only CS-based films could be tested in this way, as PEC was soluble in solutions. The film-forming suspensions of the PEC samples were therefore mixed with buffers and the color changes were recorded directly from the solutions. As expected, the color of films without extracts did not change (data not shown). The most obvious differences were seen in CS samples with BCW, as also shown previously [26,27]. It was suggested that the structural modifications of the natural pigments contained in BCW changed from a red flavylium cation at low pH (2) to a quinoidal form with a pinkish purple hue at pH 4 to 6 and to a blue anhydrous structure under alkaline conditions (pH > 8) [28]. Some differences were observed in PEC samples where a reddish color appeared under acidic conditions, changing to a pale pink or even colorless at neutral pH and changing to a reddish brown at alkaline pH. PEC films exhibited positive *a** and *b** values, while CS exhibited a positive or near zero *a** value and negative *b**, which decreased even further with increasing pH. Similarly, some authors studied chitosan/starch/anthocyanin films and chitosan/pectin films [29,30,31]. They found that the color of the samples changed to blue-green in the range of pH from 8 to 10 in the presence of anthocyanins. It can be seen from Figure 2 and Appendix A that the significant changes in the color parameters (*a** and *b**) occurred after the films were exposed to different pH buffers, indicating that the color of the produced films depended on the pH of the environment. Moreover, the intensity of color changes in the films increased with a higher concentration of BCW extract. The results indicate that blackcurrant pomace could be used as a visual indicator of pH changes if incorporated into a smart packaging material. This could be a quick and non-invasive sensor of food deterioration, such as indicating the onset of fish spoilage, which is known to release volatile amines that are responsible for pH changes.

### 2.4. Thermal Properties

The thermal behavior of CS, PEC and blends with 20% BCW is shown in Figure 3 and Table 4. CS-based films showed the first endothermic peak at 82.7 °C, which was due to the loss of moisture and traces of solvents. The second endothermic peak was found at 162.0 °C and was assigned as the dissociation temperature (*T_DS_*). Heat fusion values (Table 4) were lower in films with BCW, thus indicating a decrease in the crystallinity of samples. This was probably due to the fact that the presence of BCW did not allow for the formation of specific crystallization centers or prevent the growing of crystallites. The *T_DS_* is defined as a range where the dissociation process of hydrogen bonds between the polymer chains of CS takes place [18]. An endothermic peak (melting, *T_m_*) at 288.8 °C was attributed to further degradation, resulting from depolymerization and pyrolytic decomposition of the polysaccharide backbone. The *T_DS_* and *T_m_* values of the CS20BCW films were lower than those of the pure CS. 

It was reported, in Reference [32], that the thermal behavior of PEC mainly depends on the chemical composition and the state transitions occurring during processing. In the present study, the PEC-based films exhibited two intense peaks. The first endothermic peak at 169.7 °C for PEC and 135.7 °C for PEC20BCW was attributed to the melting of pectin crystals. A similar melting temperature (154 °C) was reported in Reference [33]. There was a similar decreasing behavior of the heat of fusion in the chitosan-based samples, also indicating a crystallinity drop. The second exothermic transition peaks at 230.2 °C for PEC and 225.8 °C for PEC20BCW were attributed to the decomposition temperature of pectin [34]. Lower values for film formulations with BCW indicate matrix decomposition and lower thermal stability of the films produced.

### 2.5. Antioxidant Film Properties

The total phenolic content, TPC, and antioxidant activity, AOA, of the tested films are shown in Table 5. BCW was found to have a considerably high content of polyphenols. The TPC in CS and PEC films increased significantly with the addition of BCW. The PEC samples showed significantly higher TPC values compared to the CS films. This is probably due to the changes in the structure of polyphenols and their binding to CS. The differences in solubility of these two polymers, e.g., during the extraction step, may affect the TPC. Thus, it is possible that polyphenols remained bound to the polymer during extraction and were not sufficiently extracted by the solvent. Knowledge of TPC in dry films shows how the antioxidant capacity of films is related to their efficacy when used as functional packaging for oxygen sensitive foods [35,36]. The AOA shows the scavenging activity of polyphenols present in BCW samples. With the increasing of BCW concentration, the AOA also increased significantly and proportionally. According to Reference [37], polymers may possess intrinsic antioxidant activity due to the remaining free groups (such as amino-NH_2_ in CS) that could absorb hydrogen ions from the solution (amination). By replacing the hydroxyl groups in polymers, or in other words, by linking them with free groups and forming the free radical end, it is possible to increase the AOA of films [38]. The AOA of BCW is attributed to its polyphenolic profile. According to the literature, BCW mainly contains delphinidin-3-*O*-glucoside, delphinidin-3-*O*-rutinoside, cyanidin-3-*O*-glucoside, cyanidin-3-*O*-rutinoside, malvidin-3-*O*-rutinoside and cyanidin-3-*O*-malonyl-glucoside, which are responsible for the AOA [39,40].

## 3. Materials and Methods

### 3.1. Materials and Reagents

Chitosan (CS) (France Chitin, Orange, FR, type 652, Mw 165 kDa, DA > 85%) and pectin (PEC) powder (citrus 121 grade, CAS 9000-69-5, Fisher Scientific, Leicestershire, UK) were used for film preparation. Blackcurrant fruit (BC) (*Ribes nigrum*) was purchased frozen, from a local supermarket, packed in polyethylene bags and stored at −18 °C. The BC was pressed and the fruit pomace was used for the preparation of active powder (BCW). Acetic acid (glacial 100%, Merck, Darmstadt, Germany), pure ethanol (96%, Gram mol, Zagreb, Croatia), deionized water and glycerol (Fluka Chemical, 98% purity, Neu Ulm, Germany) were used for film forming solutions (FFS). Commercial pH buffers (KEFO, Sisak, Croatia) were used for the evaluation of pH sensor properties. All chemicals were used as received. 

### 3.2. Preparation of Blackcurrant Powder 

BC pomace was lyophilized (frozen samples at −80 °C, lyophilization during 48 h) (ScanCool SCL210P, Labo-GeneTM, Lynge, Denmark). Lyophilized material was ground using a household blender. Particles that passed through a sieve with a nominal mesh aperture of 180 microns were collected and kept refrigerated before being used for film preparation. 

### 3.3. Film Preparation 

CS and PEC powders were dissolved in 1% (*v*/*v*) aqueous acetic acid solution and distilled water to obtain 2 and 3% (*w*/*v*) suspensions (FFS), respectively. These solutions were continuously stirred for 2 h at room temperature (23 ± 2 °C), until all the powder was dissolved. Then, 30% (*w*/*w* of solution) glycerol was added to the FFS for 30 min with stirring at 1200 rpm. Blackcurrant waste powder (BCW) was added to the FFS (at concentrations of 10 and 20%, *w*/*w*) with stirring at 1200 rpm. The enriched film forming dispersion was optimized for 2 h. The pH for CS and PEC solutions were 4.60 and 4.20, respectively. An exact amount of FFS (20 g) was then poured into a glass Petri dish (20 cm diameter) and dried for 24 h (Memmert HPP110, Memmert, Buechenbach, Germany) at 30 °C and 50% RH. The dry films were peeled off and conditioned in a controlled atmosphere at 50% RH and 25 °C until used.

### 3.4. Physical and Chemical Characterization of Films

#### 3.4.1. Film Thickness

The thickness of all film samples was measured with a digital micrometer with an accuracy of 1 µm (Digimet, HP, Helios Preisser, Gammertingen, Germany). An average of 10 values was used for further calculations. 

#### 3.4.2. Water Vapor Permeability (WVP) Measurements 

The WVP of films was measured gravimetrically ([41] adapted to edible materials by [42]). Differential relative humidity (RH) (ΔRH70) was used by adding the distilled water (100% RH) to the permeation cell (cup) and exposing it to 30% RH in a ventilated climatic chamber (Memmert HPP110, Memmert, Buechenbach, Germany), at 25 ± 1 °C. The water vapor permeability, WVP (g m^−1^ s^−1^ Pa^−1^) was calculated at the steady state of the permeation process, i.e., at a constant weight change of the cell, according to Equation (1):(1)$$\mathrm{WVP}=\frac{\Delta m}{\Delta t\cdot \;A\cdot \;\Delta p}\cdot x$$
where Δm/Δt is the weight of moisture loss per unit of time (g s^−1^), A is the film area exposed to the moisture transfer (9.08 × 10^−4^ m^2^), x is the film thickness (m), and Δp is the water vapor pressure difference between the two sides of the film (Pa). 

For all the samples, 5 replicates were performed.

#### 3.4.3. Moisture Content and Water Solubility

The dry matter of the film was determined by weighing the film of the exact area (2 cm × 2 cm) before (w_i_) and after (w_f_) 2 h drying in an oven at 105 °C. The solubility in water (WS) was evaluated according to the literature [16,43]. WS gives the information about the amount of dry solids dissolved in distilled water after the films were immersed for 24 h. The procedure was as follows: (1) cutting the film samples into discs (2 cm diameter); (2) determining the initial dry weight (w_i_) by drying to a constant weight at 105 °C; (3) immersing in distilled water (30 mL) and leaving for 24 h at 25 °C with periodical shaking; (4) oven drying (105 °C) the remaining film pieces to constant weight (the final weight of dry matter not dissolved in water (w_f_). 

WS (%) was calculated using the following equation: (2)$$\mathrm{WS}\;\left(\%\right)=\frac{\mathrm{Wi}-\mathrm{Wf}}{\mathrm{Wi}}\Delta 100$$

Three measurements were taken for each film.

#### 3.4.4. Mechanical Properties

A universal tensile testing machine (Stable Micro Systems Texture Analyzer TA.HD. plus, Surrey, UK) was used to determine the tensile strength (TS, MPa), the Young’s modulus (YM, MPa) and the percentage of elongation at a breakpoint (E, %) according to ASTM D882 [44]. The film specimens were cut into a rectangular shape (1.5 × 5 cm). Prior to testing, all samples were equilibrated for 7 days at 50% of RH and at 25 °C. The equilibrated film specimens were clamped in the extension grips of the testing machine and unaxially stretched at a rate of 50 mm min^−1^ until rupture. The initial distance between specimen holders was 4 cm. TS, YM, and E were plotted by computer from the stress–strain curves. Three replicates were tested for each film.

#### 3.4.5. Film Color

Film color was measured with a colorimeter (Chroma meter CR-5, Konica Minolta, Tokyo, Japan) using the CIE-Lab color scale. The following color parameters were measured: *L** (lightness), *a** (redness) and *b** (yellowness). From the recorded parameters *L**, *a** and *b**, the total color difference Δ*E* was determined according to Equation (3). It is defined that when Δ*E* is more than 3, the color change can be perceived by human eye.
(3)$$\Delta E=\sqrt{{\left(L_{1}-L_{0}{)}^{2}+(a_{1}-a_{0}\right)}^{2}+{\left(b_{1}-b_{0}\right)}^{2}}$$
with *L*_1_, *a*_1_ and *b*_1_—for active film, *L*_0_, *a*_0_ and *b*_0_—control film (CS or PEC without active substances). 

To evaluate the color changes of the pH sensing films, the above procedure was also used. Before measurements, the film samples were immersed in different pH buffers (pH 2, 4, 5, 6, 7, 8, 10, and 12) and left for 10 min. After that, all samples were collected and evaluated. All color measurements were taken at 5 different locations on the surface of each film sample. 

#### 3.4.6. FTIR Analysis

Fourier transform infrared (FTIR) spectroscopy was carried out using an FTIR spectrometer (PerkinElmer Frontier, Llantrisant, UK). FTIR spectra were recorded in the frequency range from 4000 to 400 cm^−1^ using ATR (attenuated total reflectance) with a ZnSe crystal. For each measurement, 64 scans were taken with a resolution of 4 cm^−1^. The spectra were recorded in duplicate. The aim of this analysis was to determine the molecular level modifications induced by the incorporation of BCW into the polymer chains.

#### 3.4.7. Differential Scanning Calorimetry (DSC) Analysis

Thermal behavior was checked by differential scanning calorimetry (DSC), using Mettler Toledo DSC822e calibrated with indium (Mettler Toledo, Greifensee, Switzerland). The film samples were preconditioned at 53% RH in a climatic chamber for at least 48 h before analysis. About 10 mg of the studied film samples were hermetically sealed in an aluminum pan and heated (two runs: 1st from 0 to 220 °C and 2nd to 350 °C) at a rate of 10 °C min^−1^. An empty aluminum pan was used as a reference. The peak points of specific temperature and heats of fusion were calculated from DSC thermograms using TA Universal Analysis Software (New Castle, DE, USA). According to the author’s previous experiences [18] and preliminary results (not shown), it was expected that the greatest changes would be observed after incorporation of the highest extract concentrations, 20% BCW in this study, therefore only these film formulations were tested. 

### 3.5. Antioxidant Properties 

#### 3.5.1. Total Phenolic Content 

Total phenolic content (TPC) was determined using the modified Folin-Ciocalteu method [45]. First, 100 µL of the appropriately diluted content was mixed with 200 µL of Folin–Ciocalteu reagent and 2 mL of distilled water. Then, 1 mL of 20% Na_2_CO_3_ (*w*/*v*) was added and kept at 50 °C for 25 min. The absorbance was measured at 765 nm using a spectrophotometer (model UV 1600PC; VWR International, Leuven, Belgium). The solvent (distilled water for PEC samples and aqueous acetic acid (1%, *v*/*v*) for CS samples) for extraction was used instead of fruit extract for blanks, and gallic acid (5 g L^−1^) was used for the analytical curve. TPC was expressed as mg of gallic acid equivalent (GAE) g^−1^ of prepared film or mg of GAE g^−1^ of powdered extract. All measurements were performed in duplicate.

#### 3.5.2. Antioxidant Activity of the Films

The reducing capacity (antioxidant activity, AOA) was determined using a modified ferric reducing antioxidant power (FRAP) method [45]. The procedure was as follows: freshly prepared FRAP reagent was mixed with 0.3 M acetate buffer (pH = 3.6), 2,4,6-tripyridyl-s-triazine (TPTZ) solution and 20 mM FeCl_3_·6H_2_O solution, respectively, in a ratio of 10:1:1 (*v*/*v*/*v*) and incubated at 37 °C. The sample (film or extract) and FRAP reagent were mixed (300 µL and 2.25 mL, respectively) and incubated for 10 min at 37 °C. The absorbance was measured at 593 nm. The blank sample was prepared using either distilled water for extracts and PEC-based samples or aqueous acetic acid for CS-based samples. The obtained results were expressed as the mg of ascorbic acid equivalents (AAE) g^−1^ of the prepared films, or mg AAE g^−1^ of the extract [46]. All measurements were carried out in duplicate.

### 3.6. Statistical Analysis

Statistical analysis was performed using Xlstat-Pro (win) 7.5.3. (Addinsoft, New York, NY, USA). All data were ranked and the statistical differences were evaluated on the ranks using one-way analysis of variance (ANOVA) and Tukey’s multiple comparison tests. In all cases, a value of *p* < 0.05 confidence level is considered significant.

## 4. Conclusions

Novel formulations of bio-based films were successfully prepared using blackcurrant waste. Blackcurrant waste was used with the aim of obtaining both antioxidant properties and colored polyphenols to be used as color indicators of pH changes. Both study objectives were achieved to an extent that can be used for further investigation and use of the films in laboratory scale tests on real foods sensitive to oxidative stress or production of spoilage-indicating volatiles that can alter the pH of the packaging environment. Significant color changes were observed in dry films as a function of pH, providing a promising mechanism for their use as intelligent sensors for food packaging. However, further analysis and improvements need to be carried out in large-scale production. The antioxidant activity was dependent on the availability of phenols entrapped in the polymer matrix, which were derived from blackcurrant waste. The incorporation of blackcurrant waste affected some physical and chemical film properties. The water vapor permeability was significantly increased after the incorporation of the active powder, which was mainly due to the physical failures in the film matrix. The mechanical efficiency, as an integral film, was also decreased due to the incorporation of undissolved particles present in the blackcurrant waste. Spectral analysis of the film surface revealed no detectable binding between active powder and polymer chains on the film surface. However, the degradation temperatures determined by DSC were lower in the samples with blackcurrant powder, indicating a disturbance in polymer stability. A decrease of the melting temperature is related to a loss of thermal stability. The developed films are a good example of the use of food waste to produce packaging materials for environmentally conscious manufacturers. Produced composites are of great interest as they are created from abundant food by-products which, on the one hand, create environmental problem, and on the other hand, are rich in active compounds. Blackcurrant has not been widely used for the production of functional products, as are some other fruits of the berry group (such as cranberry, blueberry, etc.). Therefore, additional knowledge about its potential use is of scientific and industrial relevance.

## Figures and Tables

**Figure 1 molecules-26-02569-f001:**
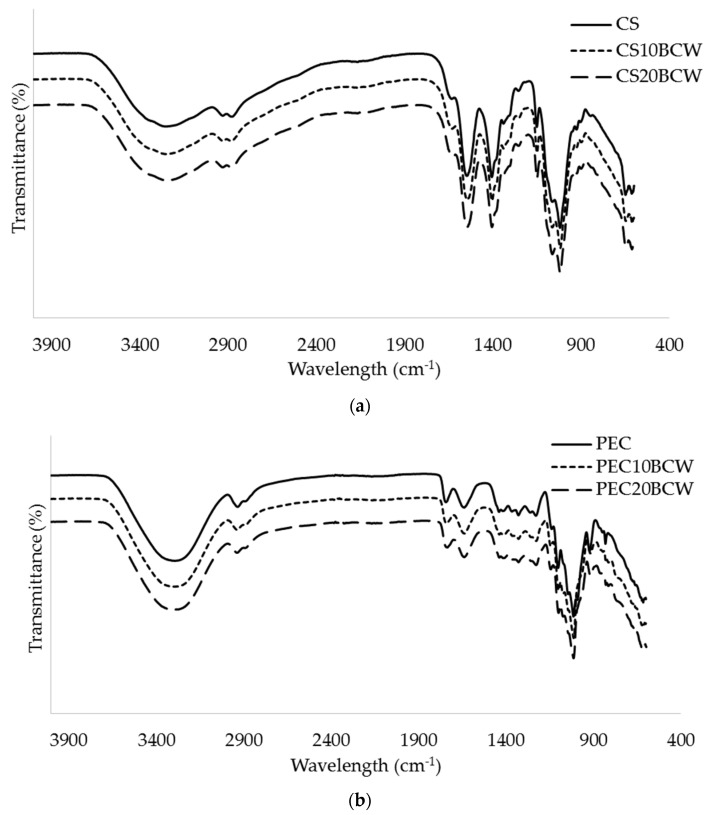
FTIR spectra of chitosan (**a**) and pectin (**b**) without or with blackcurrant powder (BCW). CS-chitosan, CS10BCW and CS20BCW-chitosan with 10 and 20% (*w*/*w*) blackcurrant powder, PEC-pectin, PEC10BCW and PEC20BCW-pectin with 10 and 20% (*w*/*w*) blackcurrant powder.

**Figure 2 molecules-26-02569-f002:**
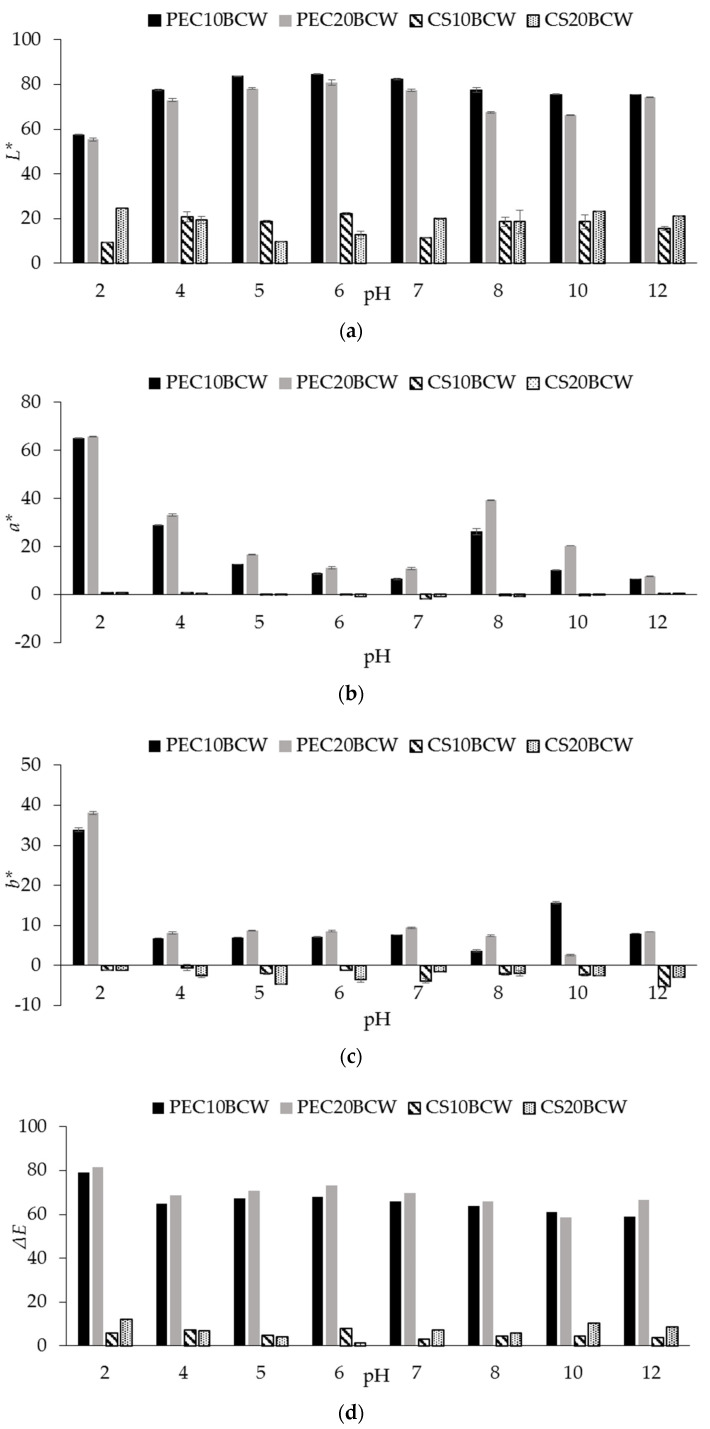
Color parameters of tested film formulations during exposure to various pH ranging from 2 to 12, given as LAB scale values: (**a**) *L**; (**b**) *a**; (**c**) *b** and (**d**) Δ*E*.

**Figure 3 molecules-26-02569-f003:**
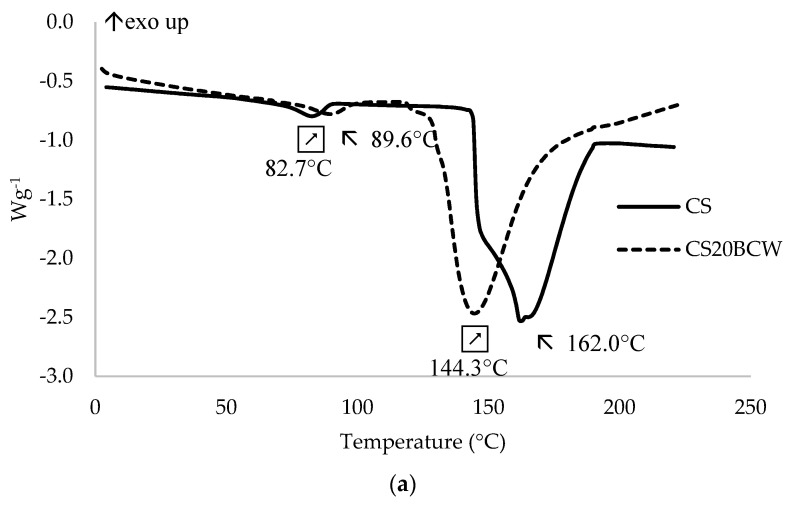
DSC thermograms (1st heating) of chitosan (**a**) and pectin (**b**) films without or with 20% (*w*/*w*) blackcurrant powder (BCW).

**Table 1 molecules-26-02569-t001:** Thickness, water vapor permeability (WVP), moisture content (MC) and solubility in water (WS) of CS and PEC films with or without blackcurrant powder at different concentrations.

Sample	Thickness (µm)	WVP (× 10^−10^ g m^−1^ s^−1^ Pa^−1^)	MC (%)	WS (%)
CS	56.71 ± 1.93 ^c^	2.84 ± 0.13 ^c^	24.96 ± 0.04 ^a^	31.20 ± 1.20 ^b^
CS10BCW	47.83 ± 3.32 ^d^	2.43 ± 0.07 ^d^	22.13 ± 4.75 ^a^	35.68 ± 0.40 ^b^
CS20BCW	62.55 ± 3.79 ^c^	2.82 ± 0.05 ^c^	19.48 ± 0.62 ^a^	30.28 ± 1.25 ^b^
PEC	59.75 ± 2.29 ^c^	2.96 ± 0.08 ^c^	16.10 ± 0.15 ^b^	100.00 ± 0.00 ^a^
PEC10BCW	113.57 ± 3.20 ^b^	3.97 ± 0.04 ^a^	15.07 ± 1.19 ^b^	100.00 ± 0.00 ^a^
PEC20BCW	145.00 ± 4.99 ^a^	3.45 ± 0.09 ^b^	22.17 ± 1.03 ^a^	100.00 ± 0.00 ^a^

Values are given as average ± standard error. CS-chitosan, CS10BCW and CS20BCW-chitosan with 10 and 20% (*w*/*w*) blackcurrant powder, PEC-pectin. PEC10BCW and PEC20BCW-pectin with 10 and 20% (*w*/*w*) blackcurrant powder. Different superscripts (a–d) within the same column indicate significant differences among samples (*p* < 0.05).

**Table 2 molecules-26-02569-t002:** Mechanical properties of the tested film formulations expressed as elongation at break (E), tensile strength (TS) and Young’s modulus (YM).

Sample	E (%)	TS (MPa)	YM (MPa)
CS	34.17 ± 0.01 ^a^	9.88 ± 0.02 ^b^	28.91 ± 0.05 ^c^
CS10BCW	16.61 ± 1.41 ^d^	13.32 ± 1.58 ^a^	79.62 ± 4.35 ^b^
CS20BCW	7.68 ± 0.69 ^e^	13.36 ± 1.31 ^a^	175.87 ± 15.41 ^a^
PEC	28.92 ± 0.45 ^b^	9.29 ± 0.53 ^b^	32.10 ± 1.57 ^c^
PEC10BCW	21.34 ± 1.34 ^c^	3.97 ± 0.04 ^c^	24.49 ± 4.42 ^c^
PEC20BCW	14.73 ± 1.45 ^d^	1.48 ± 0.14 ^d^	10.08 ± 0.77 ^d^

Values are given as average ± standard error. CS-chitosan, CS10BCW and CS20BCW-chitosan with 10 and 20% (*w*/*w*) blackcurrant powder, PEC-pectin. PEC10BCW and PEC20BCW-pectin with 10 and 20% (*w*/*w*) blackcurrant powder. Different superscripts (a–d) within a column indicate significant differences among samples (*p* < 0.05).

**Table 3 molecules-26-02569-t003:** Color parameters (*L**, *a** and *b** and the total color difference Δ*E*) of the tested film formulations.

Sample	*L**	*a**	*b**	Δ*E*
CS	16.77 ± 0.99 ^b,c^	−1.23 ± 0.12 ^d^	−3.05 ± 0.76 ^e^	0.00 ± 0.00 ^d^
CS10BCW	14.28 ± 2.34 ^b,c^	−1.52 ± 0.20 ^d^	−2.49 ± 0.34 ^d^	3.89 ± 1.15 ^c^
CS20BCW	12.77 ± 2.33 ^c^	−1.29 ± 0.11 ^d^	−2.57 ± 0.35 ^d^	4.30 ± 2.10 ^c^
PEC	96.66 ± 0.02 ^a^	0.07 ± 0.00 ^c^	4.98 ± 0.01 ^a^	0.00 ± 0.00 ^d^
PEC10BCW	16.93 ± 0.68 ^b^	6.61 ± 0.38 ^b^	−0.38 ± 0.32 ^b,c^	80.18 ± 0.65 ^b^
PEC20BCW	8.15 ± 0.33 ^b–d^	11.61 ± 0.20 ^a^	−0.09 ± 0.41 ^b,c^	89.40 ± 0.28 ^a^

Values are given as average ± standard error. CS-chitosan, CS10BCW and CS20BCW-chitosan with 10 and 20% (*w*/*w*) blackcurrant powder, PEC-pectin. PEC10BCW and PEC20BCW-pectin with 10 and 20% (*w*/*w*) blackcurrant powder. Δ*E* values for CSBCW and PECBCW are calculated with respect to CS and PEC without BCW. Different superscripts (a–d) within a column indicate significant differences among samples (*p* < 0.05).

**Table 4 molecules-26-02569-t004:** Melting temperatures (*T_m_*) and heats of fusion (Δ*H_f_*) of tested film formulations.

Sample	*T_m_* (°C)	Δ*H_f_* (J g^−1^)
CS	162.0	351.3
CS20BCW	144.3	291.6
PEC	169.7	246.5
PEC20BCW	135.7	146.4

CS-chitosan, PEC-pectin, CS20BCW and PEC20BCW-chitosan and pectin with 20% (*w*/*w*) blackcurrant powder.

**Table 5 molecules-26-02569-t005:** Total phenolic content (TPC) and antioxidant activity (AOA) in PEC and CS films with different amounts of BCW.

Sample	TPC (mg GAE g^−1^)	AOA (mg AEE g^−1^)
CS	0.53 ± 0.53 ^d^	0.89 ± 0.01 ^f^
CS10BCW	2.58 ± 0.15 ^c^	2.51 ± 0.17 ^e^
CS20BCW	2.12 ± 0.64 ^c^	3.02 ± 0.46 ^d^
PEC	0.33 ± 0.06 ^d^	0.79 ± 0.01 ^f^
PEC10BCW	3.49 ± 0.23 ^b,c^	8.99 ± 0.28 ^c^
PEC20BCW	6.15 ± 0.59 ^b^	24.26 ± 0.55 ^b^
BCW10%	13.18 ± 0.71 ^a^	52.85 ± 3.25 ^a^

Values are given as average ± standard error. CS-chitosan, CS10BCW and CS20BCW-chitosan with 10 and 20% (*w*/*w*) blackcurrant powder, PEC-pectin. PEC10BCW and PEC20BCW-pectin with 10 and 20% (*w*/*w*) blackcurrant powder. BCW 10% (*w*/*v*) blackcurrant powder solution. Different superscripts (a–f) within a column indicate significant differences among samples (*p* < 0.05).

## Data Availability

Data is contained within the article or Appendix A.

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
