# Peer review of "Characterization of Food Packaging Films with Blackcurrant Fruit Waste as a Source of Antioxidant and Color Sensing Intelligent Material"

_molecules, 2021, doi:10.3390/molecules26092569_

Round 1
Reviewer 1 Report
The results presented in the manuscript are interesting in terms of new materials. Nevertheless, after reading the manuscript, I noticed some important shortcomings that needed explanation and improvement.
Manuscript should be spell-checked and corrected, preferably with the help of a native speaker. Authors should also revise the work for minor formatting and editing errors.
Abstract: Page 1, line 18: Replace re-used in “Blackcurrant waste was re-used..”, as the waste was first time used for producing these materials.
Line 20 “naturally produced films” means films being produced by nature, which is not the case in this study, please replace it. Rephrase the whole sentence for a better understanding.
Abstract doesn’t reflect the results presented in the manuscript, should be modified accordingly, especially for chitosan films.
Introduction: Page 1, line 43 Some examples of “novel extraction methods” should be mentioned.
Page 2, line 43: “functional” appears 3 times in the same line.
Lines 78-79: Please mention some references for “Those kinds of films”
Line 87: Authors should be more specific with “those parameters”
Innovation brought by this study should be emphasized, clearly explaining also why these composites are needed compared to the existing packaging materials.
Page 8: DSC – Discussions over the influence of crystallinity changes induced by BCW presence over thermal properties should be introduced.
Water usually evaporates around 100 oC, so definitely 168.7 °C is too high for PEC dehydration in this case. A melting endothermic peak (Tm) at 154 °C was reported by Iijima, M. et al (2000). Phase transition of pectin with sorbed water. Carbohydrate Polymers, 41(1), 101–106. doi:10.1016/s0144-8617(99)00116-2
Which DSC run (I or II) is plotted in Figure 3?
Page 9: Materials: Particle sizes for BCW are essential to be mentioned, greatly influencing the films properties.
Conclusions: Page 12 Lines 439-440: This sentence is a nonsense “degradation temperatures as determined with DSC were lower in samples with blackcurrant powder indicating a disruption in the polymer stability”, as thermal stability is improved if degradation temperature is lowered.
Reviewer 2 Report
This manuscript investigated the application of blackcurrant fruit waste on the food packaging films. This manuscript was presented well with required experiments. I have following suggestion to the authors:
- The entire manuscript should be revised for English, and many sentences need to be rephrased.
- Sections number in the entire manuscript should be corrected. For instance, section 2 is “Results and Discussion” and the subsection “Physical, chemical and mechanical characterization of films” is 3.1.
- Line 251: Reference was not cited properly in the text.
- Figure 1: Why there is no change in FTIR spectra for the addition black current waste powder in the chitosan or pectin formulations? Did the authors conducted FTIR for black current powders alone? In that case, what is the authors expectation on the FTIR for black current powders.
- What is the purpose of freeze drying the BC pomace, as it was already dried (line 302).
- Line no: 326: Does the RH mean relative humidity? Mention the expansion of RH in text.
Round 2
Reviewer 2 Report
Authors response is satisfactory.